# A qualitative study of COVID-19 vaccine intentions and mistrust in Black Americans: Recommendations for vaccine dissemination and uptake

Lu Dong[1]*, Laura M. Bogart[1], Priya Gandhi[1], James B. Aboagye[2], Samantha Ryan[3], Rosette Serwanga[4], Bisola O. Ojikutu[5,6,7]

1 RAND Corporation, Santa Monica, CA, United States of America, 2 UCLA Center for HIV Identification, Prevention, and Treatment Services, Los Angeles, CA, United States of America, 3 RAND Corporation, Pittsburgh, PA, United States of America, 4 African Immigrants Community, Boston, MA, United States of America, 5 Boston Public Health Commission, Boston, MA, United States of America, 6 Division of Global Health Equity, Brigham and Women's Hospital, Boston, MA, United States of America, 7 Infectious Disease Division, Massachusetts General Hospital, Boston, MA, United States of America

* ldong@rand.org

## Abstract

### Background

COVID-19 vaccination rates among Black Americans have been lower than White Americans and are disproportionate to their population size and COVID-19 impact. This study examined reasons for low vaccination intentions and preferred strategies to promote COVID-19 vaccination.

### Methods

Between November 2020 and March 2021, we conducted semi-structured interviews with 24 participants who expressed low vaccination intentions in a RAND American Life Panel survey; we also interviewed five stakeholders who represent organizations or subgroups in Black communities that have been highly affected by COVID-19.

### Results

Many interviewees discussed the "wait-and-see" approach, citing that more time and evidence for vaccine side effects and efficacy are needed. Perceived barriers to COVID-19 vaccination included structural barriers to access (e.g., transportation, technology) and medical mistrust (e.g., towards the vaccines themselves, the government, healthcare providers and healthcare systems, and pharmaceutical companies) stemming from historical and contemporary systematic racism against Black communities. Interviewees also discussed strategies to promote COVID-19 vaccines, including acknowledging systemic racism as the root cause for mistrust, preferred messaging content (e.g., transparent messages about side effects), modes, and access points (e.g., a variety of medical and

**Data Availability Statement:** All relevant data are within the paper (e.g., results, Tables 3 and 4). Demographic dataset for the ALP participants can

be made available upon registration from American Life Panel (https://www.rand.org/research/data/alp.html).

**Funding:** Funding for this research (awarded to LMB) was provided by gifts from RAND supporters and income from operations. The funders were not involved in the study design; data collection, analysis, and interpretation; or the preparation of this manuscript.

**Competing interests:** The authors have declared that no competing interests exist.

non-medical sites), and trusted information sources (e.g., trusted leaders, Black doctors and researchers).

## Conclusions

These insights can inform ways to improve initial and booster vaccination uptake as the COVID-19 pandemic progresses.

## Introduction

COVID-19 disproportionately affects Black Americans [1]. Compared to non-Hispanic White Americans, Black Americans were 1.1 times more likely to be diagnosed with COVID-19 diagnoses, 2.8 times more likely to be hospitalized, and twice as likely to die from COVID-19 as of July 2021 [2]. COVID-19 related disparities are further exacerbated by low vaccination rates among Black Americans, particularly at the beginning of the vaccine roll-out. Since the announcement of effective COVID-19 vaccines in November 2020, Black Americans have been less likely to receive a vaccine than White Americans. While the vaccination rate has increased from January through August 2021 for all racial/ethnic groups, Black Americans have the lowest vaccination rate (25.4% fully vaccinated) compared to White Americans (33.7%) [3], and their vaccination rate remains disproportional to their population size and COVID-19 impact in terms of cases and deaths [4]. The most recent national data published from September to November 2021 have documented increased vaccination rates among Black Americans and reduced disparities in vaccination rates between Black and White Americans [5–7]. However, younger Black adults showed lower vaccination intention for the booster dose (i.e., 30% did not intend to get a booster dose), compared to younger Hispanic and non-Hispanic White adults (i.e., 16% did not intend to get a booster dose) [6]. Although vaccine intentions and rates have increased over time in Black communities [8], vaccination rates among Black Americans were the lowest compared to other racial groups during most of 2021, suggesting insufficient efforts to increase access. News reports as well as recent research data have documented accommodation and accessibility barriers with COVID-19 vaccine access in Black and other underserved communities of color [9–12].

In addition to barriers with vaccine access, medical mistrust is a factor that can influence low COVID-19 vaccination rates among Black Americans. Medical mistrust is defined as distrust of the healthcare system, providers, and treatments and is prevalent among Black Americans for various health conditions, including in the HIV epidemic [13,14]. Evidence shows that mistrust has extended to beliefs about COVID-19 treatments and vaccines [15]. Medical mistrust is rooted in the historical mistreatment of Black Americans both within health care as well as U.S. society and institutions and is maintained by persistent discrimination, inequities, and injustices. While medical mistrust is associated with negative health-related outcomes [16], recent research has shifted toward viewing medical mistrust as an adaptive and justifiable response due to negative personal experiences or historical injustice [13,17].

In a recent review, Black Americans showed the lowest COVID-19 vaccine confidence compared to other racial/ethnic groups, and medical mistrust and past racial discrimination were significant predictors [18]. Concerns about safety, efficacy, and side effects, and exposure to myths and misinformation also predicted low vaccine confidence [18]. In a nationally representative sample of Black Americans, high mistrust of the vaccine itself (e.g., concerns about harm and side effects) and weak subjective norms for vaccination in one's close social network

were predictors of low vaccination intention, and living in an area of higher socioeconomic vulnerability was a marginally significant predictor [19].

Few qualitative studies have examined the manifestation of COVID-19 related medical mistrust in Black Americans, how it influences vaccine confidence and intentions, and how culturally responsive strategies may be developed to promote COVID-19 that focused specifically on Black communities. One study conducted focus groups in racial/ethnic minority populations, including Black communities, in LA County and identified contextual (e.g., references to mistrust from unethical research studies, accessibility and accommodation barriers, inequitable access), individual/social (e.g., inadequate exposure to trusted messengers or information, medical mistrust), and vaccine-specific influences to the vaccination decision-making process (e.g., need for vaccine evidence by subpopulation, misconceptions on vaccine development) [11]. The current study, which was conducted in the early months of the COVID-19 vaccine roll-out in the U.S., contributes additional qualitative data to fill this gap. Specifically, we examined barriers to and facilitators of COVID-19 vaccination, including mistrust and vaccine access, and elicited strategies to promote COVID-19 vaccination in Black communities. We conducted semi-structured interviews with Black individuals who said that they would not or were not willing to get vaccinated as well as with stakeholders who represent organizations or subgroups in Black communities that have been highly affected by COVID-19 (e.g., people living with HIV, sexual and gender minority individuals, communities of immigrants) between December 2020-March 2021, before COVID-19 vaccines were released to all adults in the general public. Therefore, the current study provides information about factors that are related to vaccine intentions early in the vaccine roll-out process, before most U.S. residents were eligible for vaccination and prior to wide-scale efforts within Black communities to encourage vaccination.

## Materials and methods

### Design and participants

The current study had two phases. In phase one, we conducted a web-based survey of Black Americans from November 17, 2020, to December 2, 2020. The timing of the survey was about one week after Pfizer-BioNTech and Moderna made public announcements about the high efficacy of their vaccines and prior to Food and Drug Administration Emergency Use Authorization. Participants were drawn from the RAND American Life Panel (ALP), a nationally representative internet panel of U.S. adults aged 18 and older. The ALP currently has over 3,000 active members who have been recruited from several other surveys and directly for the panel using multiple modes (e.g., in-person, telephone, and mail) and probability-based sampling methods, including address-based samples and telephone (random-digit dial) samples. Full details of the ALP's methodology are available in the technical documentation [20]. We sent the survey invitation to all 318 ALP panelists who self-identified as Black Americans, of whom 207 completed the survey (completion rate: 65%). In phase two, we selected and invited survey respondents who endorsed low vaccination intention to participate in a semi-structured qualitative interview. A total of 24 ALP participants completed semi-structured interviews from December 2020 to March 2021. We obtained online informed consent from all ALP participants and verbal assent from all qualitative interview participants prior to conducting the interviews. Detailed ALP survey methodology is previously published [21], and main quantitative survey results were published elsewhere [19]. Following standard ALP rates, the incentive for the 25-minute survey was $17, and the incentive for the 60-minute interview was $40.

We assembled a community advisory board (CAB) of eight stakeholders to provide input to the study throughout the research process, and the committee convened monthly from

November 2020 to April 2021. Stakeholders represented organizations in Black communities or subcommunities (e.g., people living with HIV, sexual and gender minority individuals, individuals from immigrant communities) that have been affected by the COVID-19 pandemic. We completed interviews with five stakeholders using the same interview protocol as with participants. Before participating in the interview, stakeholders provided verbal assent and were paid $40 for completing the 60-minute interview. RAND's Institutional Review Board approved the study design and protocol. This study is reported following the Consolidated Criteria for Reporting Qualitative Research (COREQ) reporting guidelines.

## Qualitative sample selection

Overall, 79 ALP participants were eligible and invited to participate in the interview based on their responses to vaccine intention questions in the ALP survey; 24 participants responded and completed the interview. The sample size was determined given the timeframe for interviewing, and we stopped recruitment when we reached a sufficient number. Two questions on the ALP survey assessed negative and positive vaccine intentions: (1) "If a vaccine were available to prevent COVID-19, I would not get it," with response options *strongly disagree*, *disagree*, *don't know*, *agree*, and *strongly agree;* (2) "Would you be willing to get a COVID-19 vaccine when it becomes available?' with response options *yes*, *no*, and *don't know/not sure*. Two questions were used to assess whether the responses may differ by question methodology (i.e., asking whether one would not get vaccinated vs. one's willingness to get vaccinated). For the qualitative sample selection, we first selected participants (n = 65) who showed low vaccination intentions on both questions (responded *strongly agree or agree* to question 1 AND *no* to question 2). To expand the qualitative sample, we relaxed the inclusion criteria to invite additional participants (n = 14) who showed low vaccination intentions on either question (responded *strongly agree/agree* or *don't know/not sure* to question 1 OR *no* or *don't know/not sure* to question 2).

## Interview protocol and study team

The interview protocol was developed to elicit information about: (1) personal intentions and beliefs related to COVID-19 vaccination; (2) feedback and suggestions about ways to promote COVID-19 vaccination; (3) trusted information sources and social influences of vaccination information and intention; and (4) ideas about community campaigns for COVID-19 vaccination in Black communities. We used open-ended questions that were phrased in a way that did not make any assumptions and allowed for participants to answer freely. Sample interview questions for each topic are available in Table 1. As the COVID-19 vaccines began to roll out in communities across the U.S., we included a few more sub-questions in our interviews to assess additional perceptions of information sharing and messaging strategies (e.g., describing the vaccines as mandatory, giving compensation for getting vaccination). Of those interviews conducted after these new questions were developed, five interviews inadvertently did not include these particular probes, although interviewees did more generally address information sharing and messaging strategies. The CAB and all co-authors reviewed and provided feedback on the interview guide.

Two interviewers (a mixed-race woman of African and European descent [SR] and a woman of South Asian descent [PG]) completed training provided by a White female PhD-level psychologist (LMB) and an Asian female PhD-level psychologist (LD). Both interviewers had prior qualitative research training or experience and research interests on the study topic: one was a graduate student, and one was a baccalaureate-level research assistant. They were trained with the interview protocol, including an overview of qualitative methods and

**Table 1. Sample interview questions by topics.**

| Topics | Sample Questions |
|---|---|
| Personal Intentions and Beliefs related to COVID-19 vaccination | • What, if anything, have you heard about the process of how COVID-19 vaccines are being developed?<br>• When a COVID-19 vaccine becomes available, how much would you want to get vaccinated? Why or why not?<br>• What types of things would make it more difficult or challenging for you to get vaccinated for COVID-19 (when a vaccine becomes available)? |
| Feedback about Future COVID-19 Vaccine Programs | • What challenges do you think there might be to provide a COVID-19 vaccine in your community? In Black communities in the U.S.?<br>• Where and how do you think people would be most comfortable getting a COVID-19 vaccine in Black communities?<br>• What do you think about creating online discussion groups or community events where people from your community describe their concerns about the vaccine and their experiences in getting the vaccine? |
| Social Influences | • Who or what sources would you trust most to give you information and advice/guidance in your community around a COVID-19 vaccine?<br>• Think about the people you know, such as your family, friends, partners, and acquaintances. How much do you think people you know would approve of you getting a COVID-19 vaccine? Why?<br>• Who in your social circle do you think would disapprove of you getting the COVID-19 vaccine? Why? |
| Ideas about Community Campaigns for COVID-19 Vaccines | • If we were to create a campaign to promote COVID-19 vaccination for people in Black communities, what messages would you recommend? What should we tell people about a COVID-19 vaccine? What could someone say to you to make you want to be vaccinated? What would you need to know about the vaccine to make you want to be vaccinated?<br>• How would you react to hearing testimonials about Black people who got vaccinated or stories of Black people who got COVID-19?<br>• What other ideas do you have for promoting the COVID-19 vaccine to people in Black communities? |

interview techniques, and completed two mock interviews each and received feedback on two practice interviews each. Interviewers completed each interview independently (i.e., the interview included only the interviewer and the interviewee). Interviewers and ALP participants had no prior relationship, but stakeholder participants knew both interviewers and the study team from ongoing CAB meetings for this study. All interviews were completed in one setting (i.e., no repeated interviews) via phone and lasted about 60 minutes each. Interviewers took detailed notes during the interview and audio-recorded the interview for transcription. Transcripts were not returned to any participant for comments, while stakeholder participants provided feedback on the qualitative findings during regular CAB meetings.

## Data analysis

Sociodemographic information from the ALP survey was linked to the interview narratives. We used Stata 17 to obtain basic descriptive statistics of the qualitative interviewees. All qualitative interviews were audio-recorded and transcribed verbatim. Qualitative data analysis was conducted in Dedoose. We followed standard procedures for direct content analysis [22], in which we developed the initial codebook based on existing research and from directly interpreting the meaning of the interview transcripts. Specifically, two experienced team members (LMB and LD) read all transcripts and developed a draft codebook based on the research questions, relevant literature, and the interpretation of the narratives. Two coders (LD and PG) coded six transcripts to test the codebook and make necessary updates through iterations. Codes were generally in three groups: vaccination intentions, barriers to vaccination, and

facilitators to (i.e., system-level strategies to improve) vaccination. The coding categories were based on the semi-structured interview guide and domains of interest. Two interviews were used to establish inter-coder reliability. The average Cohen's kappa across all codes was 0.80 ($SD$ = 0.21, range 0.30–1.00), the average prevalence and bias-adjusted kappa (PABAK) was 0.96 ($SD$ = 0.41, range 0.89–1.00), and the average percent agreement had a mean of 98% ($SD$ = 2%, range 94%-100%). For the Cohen's kappa, two codes fell below 0.50 (0.30 and 0.49, respectively) due to the low frequency of applying these two codes in the two transcripts used for establishing reliability; hence, the two coders had additional training and discussion around these two codes before proceeding with coding all transcripts. One coder (PG) then coded all remaining transcripts and held a weekly discussion with a senior team member (LD) to resolve any coding questions. Following prior research [23], we used the following categories to indicate the relative frequency of each code: most or almost all (61%-100%), some (21%-60%), and a few (1%-20%). Given that the interviews were semi-structured and used open-ended questions, providing exact percentages for codes could be misleading as a lack of discussion of a theme did not mean agreeing or disagreeing with the idea.

## Results

### Sample characteristics

Table 2 shows the sample demographic characteristics of the ALP participants and stakeholder participants. ALP participants were on average middle-aged; the majority were female and college educated. Stakeholders were on average middle-aged, and the majority were male and working in community settings and/or health-related fields.

### COVID-19 vaccination intentions

Participants mentioned a number of drivers for lower vaccination intentions, including discussing a general distrust of health care that stemmed from the U.S. Public Health Service study on syphilis at Tuskegee, other forms of medical experimentation, and personal

**Table 2. Sociodemographic Characteristics of American Life Panel (ALP) participants and stakeholder participants ($N$ = 29).**

|  | ALP Participants ($n$ = 24) | Stakeholder Participants ($n$ = 5) |
|---|---|---|
| **Demographic** | *M (SD)* or *N (%)* | *M (range)* or *N (%)* |
| Mean age (years) | 47.46 (10.33) | 44.40 (32–60) |
| Self-reported female gender | 19 (79%) | 1 (20%) |
| Sexual minority | 3 (13%) | 2 (4%) |
| Black/African American | 24 (100%) | 5 (100%)* |
| Hispanic or Latino | 2 (8%) | 0 (0%) |
| Less than bachelor's degree | 10 (42%) | n/a |
| Currently employed | 16 (67%) | 5 (100%) |
| Annual family income |  | n/a |
| less than $35,000 | 5 (21%) |  |
| $40,009 to $99,999 | 12 (50%) |  |
| $100,000 and above | 7 (29%) |  |
| Married/living with a partner (vs. separated, divorced, single) | 6 (25%) | n/a |

*Note.* *One stakeholder identifies as sub-Saharan African.

experiences with the medical system as well as general concerns around vaccines. This mistrust was commonly exhibited in terms of a "wait and see" approach, such that participants said that they were reluctant to be "*the first one in line*" (62-year-old female; education, training and library occupation) to get vaccinated but that they might be willing to get vaccinated in the near future. Most participants discussed waiting until they had an opportunity to get additional specific information on the vaccines' efficacy and safety, both formally, through healthcare providers and scientists, as well as through observations of the physical reactions of others who had received the vaccine in terms of side effects and adverse events. The more that other people they knew received the vaccines and were not adversely affected, the more willing they would be to get vaccinated themselves. For example, participants said: "*We don't know what [the vaccine is] going to do. Let them figure it out for a little while and see what happens. Then, we'll think about it. Right now, no.*" (45-year-old female; personal care and service occupation); and "*There is nothing that is going to convince me... I'll be sitting and waiting to see what happens to everybody else, regardless of race*" (49-year-old female; community and social services occupation)

Most participants said that their attitudes had not changed since the U.S. 2020 presidential election. However, a few participants described changing attitudes following the 2020 election in several different scenarios. First, some reported a better outlook for the COVID-19 vaccine roll-out after the election. For example, one participant said: "[Because] *there's now competency leading the effort and really smart bright people who are committed to getting folks vaccinated as quickly as possible*" (57-year-old male stakeholder; sexual health/HIV prevention services). Another said: "*Since the election, I think that my attitude had changed quite sufficiently because the government at that before the election wasn't willing to do anything about it.*" (72-year-old female; healthcare support occupation).

A few participants reported a change toward a more negative view of the COVID-19 vaccines due to growing mistrust: "*Since the 2020 election, my opinion has grown even more, leery of the vaccination. And that's not due to President Trump because I proudly am a supporter of President Trump. But I just think that kind of like there's something under, underlying situations. I guess it just doesn't like they go over too well. I think there's more to it, to the vaccine, and I don't trust it.*" (41-year-old female; community and social services occupation)

Consistent with the "wait and see" approach, a few participants interviewed later in the course of this study described changing their minds due to time passing and witnessing few, if any, incidences of immediate negative reactions or short-term side effects—and, thus, increased trust in the vaccine: "*I would say that right now, people are changing their minds. So it's looking more favorable than it was a few months ago.*" (40-year-old female; healthcare support occupation); and "*I don't get vaccines in general, not specifically against COVID... Recently, since they have so many people have been getting the vaccine, people that I know, and so far they've been fine... We don't know the long-term effect, but I feel comfortable now, getting it whenever it's been made available to me*" (50-year-old female; healthcare support occupation).

## Barriers to COVID-19 vaccination

Table 3 presents illustrative quotes for each theme and subcategories for structural barriers to vaccine access and systemic racism as barriers to COVID-19 vaccination.

## Structural barriers to vaccine access

Almost all ALP participants and stakeholders discussed various types of structural barriers to vaccine access, including physical access barriers (e.g., transportation issues), financial

**Table 3. Barriers to COVID-19 vaccination from semi-structured interviews (*n* = 29).**

| Barriers | ALP Participants (*n* = 24) | Stakeholder Participants (*n* = 5) |
|---|---|---|
| **Structural Barriers** | | |
| Access barriers: physical/ transportation | "They [government] want people to drive 25 minutes to get the vaccine versus just being able to go down the street. Because a lot of people in those communities, because the lower-income community don't have transportation to get to the vaccine, so you have to bring the vaccine to them." (50-year-old female, healthcare support) | "[A challenge is] having to have a car. . .I didn't see a walk-up option yesterday where I went to get a vaccine." (32-year-old male, senior project manager) |
| Access barriers: financial | "I think that [access to healthcare or insurance would] be a big, big issue because a lot of African Americans don't have health insurance. And now with the COVID they have no job. So the relief that the government sends only partially helps, doesn't help completely." (72-year-old female, healthcare support) | "If they have to access it through a medical care facility or they have to access it through their insurance company that I think that could definitely be a barrier for a lot of folks unless it comes through [Medicaid]" (40-year-old male, coordinator at an LGBT center) |
| Access barriers: technology | "You hear on the news that people don't even have a computer to do homeschooling. Those are the people I'm thinking about right now. How do you. . .get to those people?" (51-year-old female, computer & mathematical occupation) | "The last week I had somebody who is an I.T. expert telling me how complicated it was for him to sign up to get vaccinated. . .so I am wondering, okay, so if this guy could not use the computer or phone to sign up for an appointment, how much more other people who are less knowledgeable about computers [would need to sign up]?" (60-year-old female, patient navigator at a hospital) |
| Other barriers (e.g., access to information, time away from work) | "Then there's just the availability because I work. . .most things are done during business hours, right?" (40-year-old female, health support) | "If folks knew where the vaccination sites were within their area. . .if I saw or heard more of that on the radio or people had access to more information, then that might make it easier." (40-year-old male, coordinator at an LGBT center). |
| **Systemic racism as a barrier that leads to mistrust** | | |
| Racism as a root cause of mistrust | "Experiences of racism, or just inequality period, has affected or will affect people's decision to take the vaccine or not just because when you feel like you're always being attacked or no one's listening to you or no one hears you or no one sees you all of a sudden, here's something for you, you feel a little wary about it." (45-year-old female, education, training, and library) "We don't fare well when it comes to experimentation or implementation of medication. I have family members who were directly affected from the Tuskegee experiments. . .the government doesn't really care about what happens to Black people. . ." (56-year-old male, business and financial operations) | "They don't trust it at all. They think that we're being injected. It's like Tuskegee. That's the Tuskegee eugenics. All of these different things are coming up for a lot of the community that I'm interacting with. . . generations haven't forgotten." (40-year-old male, coordinator at an LGBT center) "I don't believe Black people were the in the thought of the vaccine [development]. [Black people are] hardly ever the first thought for health things like that in the U.S." (32-year-old male, senior project manager) |
| Mistrust of the vaccine itself (lack of confidence in safety/ efficacy) | "The only thing that's kind of standing in the way. . . it's the side effects, and if the shot is actually gonna do what it's supposed to." (40-year-old female, building and grounds cleaning and maintenance) | "The disadvantages are the unknown side effects or the side effects that are not as [well-studied]" (32-year-old male, senior project manager) |
| Misinformation/myths about the COVID-19 vaccines | "I believe the vaccine was likely already being experimented or while [the scientists] were creating or studying the COVID" (52-year-old female, computer and mathematical) | "I've heard that the COVID-19 vaccine is supposed to have nano bites in it that are going to control us like zombies. I've heard that it's an experiment. I have heard that it's out to kill people in particular Black folks and Brown folks." (40-year-old male, coordinator at an LGBT center) |
| Mistrust of government | "The government doesn't really care about what happens to Black people. I'm not interested in being another guinea pig or statistic." (56-year-old male, business and financial operations) | "Even still historically in the Black community, it's still traumatized, it still hurt, it's still going to mistrust not only the government and medical physicians or medical providers, but I also think, the way that the last four years rolled out with Trump didn't help."(40-year-old male, coordinator at an LGBT center) |
| Mistrust of pharmaceutical industry | "I don't say that they are trying to exterminate Blacks, but if that's side effects of their making money, I think they would be happy with that." (56-year-old male, business and financial operations) | "When you think of pharmaceuticals or those who are putting out these vaccines or drugs. . . they're thinking about the wide-scale of how we can help the most people and not people who are affected the most" (33-year-old male, research analyst) |
| Mistrust of healthcare providers/systems | "A lot of people do not know if the person standing next to them is a racist or if your health care [provider] is a racist, you don't know they have your best interest at heart due to the color of your skin" (43-year-old female, healthcare practitioner and technical) | ". . .racism and the lack of cultural responsiveness in health care for Black communities and other communities of color definitely plays a little in the vaccine and how we engage in that system" (32-year-old male, senior project manager) |

barriers, and digital and technological barriers. As one stakeholder pointed out, "*access is going to be the number one barrier*" (32-year-old male; senior project manager). Some interviewees cited physical access barriers frequently, including a lack of adequate transportation access. Most interviewees mentioned potential financial access barriers, primarily due to a lack of health insurance coverage, if there was a cost associated with obtaining the COVID-19 vaccines. A few interviewees described access barriers related to availability, such as employment and caretaking responsibilities during standard business hours. Some interviewees mentioned access barriers related to technology, including a lack of digital access and difficulty for some population segments in navigating online systems.

## Systemic racism as a barrier that leads to mistrust

Table 3 presents illustrative quotes for systemic racism as the root cause of mistrust and each type of mistrust. Mistrust was perceived to be a key barrier to vaccination. Interviewees described different types of mistrust that underlie low confidence in the COVID-19 vaccines, including misinformation and myths about COVID-19 vaccines, mistrust of government (e.g., elected officials, government information around COVID-19), medical mistrust (of healthcare organizations, providers, and the vaccines themselves), and mistrust of pharmaceutical companies. Systemic racism was discussed as the root cause of the different types of mistrust.

While a few participants said that personal experience of discrimination might not play a role in vaccination decisions given that, as one interviewee said, "*the vaccine was not made for a particular person of a particular color*" (59-year-old male; healthcare practitioner and technical occupation), almost all pointed to historical and persistent systemic racism as the root cause of mistrust of the COVID-19 vaccines. Participants and stakeholders cited various examples of racism, mistreatment, experimentation, and injustice from the past by government, pharmaceutical companies, healthcare providers and systems, and researchers and scientists, as factors that influence Black Americans' decisions for vaccination and research (e.g., clinical trial) participation. A few participants stated that they did not want to be a "guinea pig." Some interviewees also said that mistrust in Black communities could be explained by persistent and unaddressed injustice and racism across different institutions and sectors of society. For example, one interviewee said: "*Everything basically goes together [*because Black people are]* living in a place that allows the police to brutally kill your people. You see at the hospitals they're doing the same thing.*" (46-year-old female; healthcare practitioner and technical occupation).

Participants and stakeholders noted mistrust of government as a reason behind low vaccine confidence and vaccination intention. Some said that the government and politicians were telling the truth about the COVID-19 cure and vaccines: for example, one said that "*they [the government and politicians] are controlling it. Now I believe that if there's an actual cure, maybe they said don't cure it*" (49-year-old female; community and social services occupation). They also discussed the untrustworthiness of the government for Black communities specifically, citing historical and persistent mistreatment and racism. Some mentioned specific politicians that are untrustworthy and having monetary motives for promoting the COVID-19 vaccines; "*when Trump has something to do with it, it's about the money*" (54-year-old female; community and social services occupation). Some talked about the government's role in the vaccine development process as allowing a rushed process; "*they probably rushed the development and probably were allowed to skip on some of the normal precautions that would be taken for medications to come to market.*" (56-year-old male; business and financial operations occupation).

Mistrust of pharmaceutical companies was also common. Most participants and stakeholders said that the pharmaceutical companies are profit-driven and do not have the best interests of Black people. In addition, some interviewees discussed mistrust of healthcare providers and

healthcare systems, including anticipated discrimination and negative personal experiences with healthcare providers. For example, one participant cited: "*seeing family members have procedures that didn't need to be had that resulted in either long-term side effects or death, so it's just there's a lot of mistrust, not just in the history books, but also personally*" (45-year-old female; education, training, and library occupation). Some stakeholders cited providers' lack of cultural competence, sensitivity, and responsiveness in practice as a reason for mistrust of healthcare providers and systems. Some participants discussed that they felt doctors do not have patients' best interests in mind when making healthcare decisions.

Participants and stakeholders also discussed mistrust of the vaccines themselves, reporting a lack of confidence in the safety and efficacy of the newly developed COVID-19 vaccines. Most interviewees were concerned that the complete list of side effects, particularly the long-term side effects, were not yet known. Relatedly, some participants expressed concerns about a lack of clinical trial data showing the side effects and supporting the vaccines' effectiveness due to a rushed vaccine development process. For example, one stated that "*it has not been tested for a long period of time. . . and the clinical trials were rushed*" (45-year-old female; personal care and service occupation). Some participants also voiced that they were most concerned about the efficacy of the vaccines. For example, one participant said: "*the only thing that's kind of standing in the way [of getting the vaccine], I keep saying this, it's the side effects and the shot is actually going do what it's supposed to*" (40-year-old female; building and ground cleaning and maintenance occupation). Another participant expressed concerns that "*they [doctors] don't know if this vaccine is going to work on the new strain, which seems to be more easily transmitted, not necessarily more deadly, but definitely more easily transmitted*" (56-year-old male; business and financial operations occupation).

In addition, some participants and stakeholders mentioned the presence of COVID-19 related misinformation as a driver for low vaccine confidence. For example, one said: "*family members and friends who have fear and anxiety think they heard something about somebody that heard something about somebody else, and then they're giving information based upon maybe a little bit of some facts, but most of it is from fear and anxiety of what they think or believe*" (38-year-old female; healthcare support occupation). A few participants expressed beliefs consistent with misinformation, such as the virus that causes COVID-19 disease is man-made, and the vaccines were already being made when the virus was created.

Finally, a few interviewees discussed concerns regarding privacy and security of any personal information that was mandatory for vaccination. For example, one stakeholder said: "*you have to think about the implications of me coming to [the vaccination] location. And you're asking me for my name, my address, my social. I don't know what else they ask*" (60-year-old female; patient navigator at a hospital).

## System-level intervention strategies to promote COVID-19 vaccines in black communities

Table 4 shows the overarching system-level intervention strategies recommended and the trusted and preferred information sources, contents, and modes for promoting COVID-19 vaccines in Black communities. An overall recommendation was that systemic racism and past injustices should be acknowledged as a justifiable reason for mistrust in messages that promote vaccines. Relatedly, a few interviewees emphasized the need for structural changes to address racism and stop inequities and injustices: "*America will have to address a much larger issue first, which is racism. And with that, that also calls for a conversation about inequality. And with inequality, there also comes for what is it that you can do to undo that pain and suffering that*

**Table 4. System-level interventions strategies to promote COVID-19 vaccines from semi-structured interviews (*n* = 29).**

| Domains | ALP Participants (*n* = 24) | Stakeholder Participants (*n* = 5) |
|---|---|---|
| **Overarching system-level intervention strategies** | | |
| Acknowledging systemic racism | "I think one of the ways it could be addressed is definitely would have to be a mass public service campaign. First of all, emphasizing that 'Hey, we understand why you are apprehensive. We understand that these things that happened in the past to your community and other communities. But what we want to show you now is that you will be able to get the vaccine free. In addition, you have access to any follow-up care that you might need, you have 24-hour resources, hotlines, people in the community.' People actually need to get out in the community and talk to our community." (57-year-old female, business and financial operations) | "[I] encourage [that] because it's acknowledging the reality of the world that we live in and the history." (57-year-old male, sexual health HIV prevention services) |
| Improving trustworthiness of institutions (e.g., government) | "When people start seeing changes, then they'll be able to be more susceptible to accepting a vaccine or anything. But right now it's at a standstill because nothing has changed. . . they would have to see that the government and the police forces in all these different places are doing something different. They're putting something in place, stop the people from being mistreated." (46-year-old female, business and financial operations) | "People want to know exactly what's happening. Not sugar coat it, not cookie cut it. Just be real. I think that if honesty and integrity were more a part of how the government runs, this problem [of vaccine hesitancy] wouldn't be as big as it is." (40-year-old male, coordinator at an LGBT center) |
| Tailored approach | ". . . we don't all think alike, we are not all bunched up. We are all not thugs, we are not ignorant. Just like I'm gonna say for Black folks, we are all not the same. . . when you have a group of people that have been through so much in our lives, we are leery of everything, especially when it comes to living or dying." (54-year-old female, community and social services) | "There's a lot of factors I think that goes into if someone in the Black community, which is not a monolith, so if specific sects take it, or different parts of the community get it when their turn comes." (40-year-old male, coordinator at an LGBT center) |
| Use vaccination as an opportunity for healthcare and social service engagement | "Just having the resources readily available without having to turn anyone away. . .being able to service the community. . . for the homeless community like a free meal or something like that" (41-year-old female, community and social services) | ". . .to use vaccination as an opportunity. If people aren't signed up for health care to get them into health care. So people see a win, win us, and you get the free vaccine and sign up for the ACA at the same time" (57-year-old male, sexual health HIV prevention services) |
| **Preferred messaging contents** | | |
| Transparent/honest messaging | ". . .just having somebody who is fully knowledge[able] about the vaccine and that is not going to withhold information. . .Just be completely transparent and [we] will be more safe, feel more safer [and] trusting of what you think. And then show us more research." (45-year-old female, personal care and service) | "I think it's important to have accurate information, really clear messaging about the efficacy of the vaccine: why it's important, where you can get it, that there's no cost, all [of] those things that people go a long way to, uh, helping Black folks access to that vaccine." (57-year-old male, sexual health HIV prevention services) |
| Detailed information in accessible language | "Clear, concise and messages that are not so what can I say. . .in more layman's terms. Because a lot of the information that you get from, by the companies and stuff like that, just things that people just don't really understand sometimes." (45-year-old female, personal care and service) | "So give it to me in, in layman's terms, we're not doctors. And then, make it accessible. . .But like short, sweet, concise, and something that, I think that because that's, that's mostly what I'm getting from talking to people." (40-year-old male, coordinator at an LGBT center) |
| Research findings (e.g., side effects, long-term effects) | "I would want to hear the long-term impact, the research that has been done, the true facts of what is in the vaccine and how it will impact in the going forward." (42-year-old female, computer and mathematical) | "Being honest and true about [the efficacy] and what that really means for people. So people see the 95 or the 77 they're like, 'Oh, I don't want to. It's only 77% effective. . .'" Like explaining what that means. (32-year-old male, senior project manager) |
| **Trusted information sources** | | |
| Black scientists/ researchers | "If they see that maybe like a prominent Black figure is at the core front of this experiment, then the Black community, we'll see it as like a sign of greatness. So 'Oh wow, if this Black scientist had a hand in creating this history-making vaccination, then it really has to be good for us.' There's no wrong with it because a Black woman helped, and she's trying to help her community. . ." (41-year-old female, community and social services) | "If that doctor [Dr. Corbett] sat and talked to community folks or sat in and had interviews that were recorded with celebrities or what not, but had that conversation, that dialogue, I think that a lot of our communities could understand that that [the vaccine] was developed by one of our own." (40-year-old male, coordinator at an LGBT center) |

(*Continued*)

**Table 4.** (Continued)

| Domains | ALP Participants (*n* = 24) | Stakeholder Participants (*n* = 5) |
|---|---|---|
| Known, trusted figures (e.g., Black celebrities) | "If you have people trusted in the Black community actually taking the vaccine, that would be a powerful nexus tool" (57-year-old female, business and financial operation). | "A lot of these pop culture people, the celebrities, they would be great to kind of lead some of these messages." (32-year-old male, senior project manager) |
| Local community leaders | "More of the leaders in Black communities come out and get vaccinated. More of them show out and say to tell us that 'this is a process that I went through. This is how I feel after I got it. . . This is how you can go and get it. And this is how you may feel after you get it." (32-year-old female, healthcare practitioner and technical) | "I think also the messengers need to be, folks need to be Black folks in the community and not just one celebrity type person. The pastor and local politicians and teachers and people that are respected in community who are Black, who can speak to their folks, so there needs to be a variety of messengers." (57-year-old male, sexual health HIV prevention services) |
| Trusted doctors/researchers | "Regardless of who you are, you have questions. And someone from the medical field, the science field, would be better equipped to answer those questions instead of giving their, instead of giving their personal beliefs or biblical beliefs." (56-year-old female, education, training and library) | "Personally, I would, I want Fauci to call me and let me know like what's going on. If Dr Fauci could call myself on and be like, "Hey, [retracted], check it out. This is what the deal is." Woo, yeah, I would definitely, like, get it from Fauci." (40-year-old male, coordinator at an LGBT center) |
| **Preferred vaccine access points** | | |
| Having a variety of access points | "More the merrier. That's more outlets for people to get vaccinated because some people don't like to go to hospitals, but they feel better getting it in the church or in the supermarket or in the pharmacy rather than the hospital itself." (49-year-old female, community and social service) | ". . .places that feel safe for them. . .it could be a library, [a] church, it could be a community center. Can be a local, community clinic that people really see kind of set up on a big hospital chain or whatever. A mobile unit, it can be places like that. There needs to be a variety of ways that people are able to access the vaccine, not just one size fits all." (57-year-old male, sexual health HIV prevention services) |
| Medical facilities | "You feel more comfortable [getting the vaccine at a hospital], so I can monitor as I'm back and forth into my doctor's office and see how you're reacting." (45-year-old female, personal care and service) | "Wherever they are already receiving care, if they are receiving care, would be [where to get the vaccine] because they were already kind of creating that relationship, whether it be their PCP or sometimes some other specialty provider." (33-year-old male, research analyst) |
| Non-medical facilities: community organizations & faith-based organizations | ". . .probably better received because the organizations that's outside of healthcare they are geared towards helping [individuals]. They offer like food and different things that they need. So they're more trusting in that arena as opposed to the health care arena." (46-year-old female, Healthcare practitioner and technical)<br>"For some people, [faith-based organizations] would work because whatever the pastor says, they're gonna do." (56-year-old female, education, training, and library) | "I would say places like places of worship. . . going to a church or a place where community members convene, and they're being invited by a community member they know, or community members are going to participate in the process of vaccinations, somehow. I think that people will show up and be vaccinated." (60-year-old female, patient navigator at a hospital) |
| Non-medical facilities: convenient locations (e.g., local pharmacies, drive-throughs) | "That's more outlets for people to get vaccinated because some people don't like to go to hospitals, but they feel better getting it in the church or in the supermarket or in the pharmacy rather than the hospital itself." (49-year-old female, community and social services) | "Another site I went to, we were able to drive up. There's no drama whatsoever, so it's very easy process. So I think making it as easy as possible for people, is something that it's like a decision in my mind." (57-year-old male, sexual health HIV prevention services) |
| **Preferred mode of information** | | |
| Online webinar/ forum/ Q&A | ". . .while it was a webinar, they did have Q&A at the end. And they did encourage people to ask questions. And that is the only way that I think people will start to have some trust." (45-year-old female, education, training, and library) | "I think that it would give people a voice to kind of say how they feel, but also would give them that buy-in. . .I think that should be on social media or maybe Zoom. . ..I think it should be accessible to everyone. . . So I think that's the only thing is just that different people having to agree that they can disagree." (40-year-old male, coordinator at an LGBT center) |
| In-person promotion | ". . .just set up in their communities. And if necessary, go door-to-door." (62-year-old female, education training, and library) | |
| News media outlets (e.g., T.V. network) | "[On what can support the vaccine roll-out] probably a lot of advertising on local radio channels. Local news channel. . .. Just really advertisement, education, getting the word out." (50-year-old female, healthcare practitioner/technical) | "I think in terms of news sources, probably MSNBC like a Rachel Maddow, or Tiffany Cross or Joy Reid."(57-year-old male, sexual health HIV prevention services) |
| **Additional specific system-level intervention strategies** | | |

(*Continued*)

**Table 4.** (Continued)

| Domains | ALP Participants (*n* = 24) | Stakeholder Participants (*n* = 5) |
|---------|------------------------------|--------------------------------------|
| Describe vaccination as empowerment | "…makes a person feel like they're included and that they are helping the community to be better by [being vaccinated]. And you feel like you had a hand in helping decrease the spread of COVID-19 in your community. It gives a positive feeling." (38-year-old female, healthcare support) | "I think [vaccination] speaks to the power of the individuals to protect themselves" (57-year-old male, sexual health HIV prevention services) |
| Discuss vaccination as a choice | "All Black people are not the same. Everybody has their own individual mind." (51-year-old female, computer and mathematical) | "…it puts the choice in the persona's hand, but then it also brings (the choice) back to their communities…I like that" (32-year-old male, senior project manager) |
| Incentives may have limited appeal | "If there is a way to financially compensate us for going to get vaccinated, a lot of the Black community will go 9 times out of 10." (32-year-old female, healthcare practitioner and technical) | "…there could be a dinner when you get it in the grocery stores and things that will incentivize people for vaccines… when you get the vaccine, I think that [a little gift card] would be a great incentive for Black Communities" (32-year-old male, senior project manager) |

*the Black community has built for countless generations.*" (29-year-old male; management occupation).

These structural changes are needed to increase the trustworthiness of the government in Black communities, which ultimately influence vaccine acceptance: "*When people start seeing changes, then they'll be able to be more susceptible to accepting a vaccine or anything.*" (46-year-old female; healthcare practitioner and technical occupation). In addition, interviewees discussed that the Black community is not a monolith. Therefore, it is crucial to tailor the messages in terms of the content, information source, and mode for the specific community, based on research to understand reasons behind low vaccine confidence and to identify trusted sources of information and preferred mode in each community. Taking a one-size-fits-all approach is insufficient: for example, one stakeholder said: "*there needs to be a variety of ways that people are able to access the vaccine, not just one size fits all*" (57-year-old male, sexual health HIV prevention services).

Interviewees said that the content of the vaccine messages and communication strategies to promote vaccination should address the main reasons for low vaccine confidence and vaccination intention, including concerns about harm and side effects, with specific information and data from clinical trials. Participants expressed interest in hearing honest and transparent messages about the vaccines, including both good and bad outcomes from the trial, and a more detailed and well-rounded discussion about the research results in layperson terms, to increase trust and reduce concerns around the vaccines. For example, one interviewee said: "*…just having somebody who is fully knowledge about the vaccine and that is not going to withhold information…Just be completely transparent and [we] will be more safe, feel more safer [and] trusting of what you think. And then show us more research*" (45-year-old female; personal care and service occupation). Some participants suggested promoting Black scientists' involvement in the vaccine development process, mentioning Dr. Kizzmekia Corbett at the National Institutes of Health, who helped to develop the Moderna COVID-19 vaccine.

Interviewees suggested several trusted information sources to promote trust in the vaccines and encourage broader trust in clinical trials, including the need to hear from Black healthcare providers and Black scientists involved in the vaccine development process (such as Dr. Corbett) and influencers in Black communities (e.g., celebrities) as well as testimonials by Black community members or trusted leaders who share their vaccination experience. For example, one interviewee said: "*… Black representation is present in the creation of the vaccination. I hear that one of the scientists that was partially responsible for creating or working on the vaccine was a Black descent or African American herself. So I think that if Black people are aware of that,*

*they are more prone to or more willing to go along with the vaccination*" (41-year-old female; community and social services occupation). Those perceived to have ulterior motives are perceived as not trustworthy.

Interviewees made suggestions for the preferred modes of information, including online presentations, Q & A sessions, and community forums. As one interviewee said: "*for the younger generation, social media mixed with a face that they know. That means teaming up with like, the Cardi B's of the world. . .you have a local face that's well known and disseminating information. . .*" (40-year-old female; healthcare support occupation). Participants also said that it is crucial for these sessions to be led by healthcare providers, especially Black healthcare providers, and that the sessions contain opportunities for dialogue and questions to be answered. Participants also suggested that providing in-person promotion (e.g., going door to door) and using print material can help to address access barriers related to technology and digital platforms.

Regarding preferred vaccine access points, interviewees said that there needs to be vaccination access across a diverse array of medical facilities and non-medical community organizations and settings. Interviewees were mixed in preferences for places to access the vaccines (e.g., medical organizations, including hospitals and clinics, and non-medical, community-based organizations like community centers and faith-based organizations). They emphasized the importance of offering vaccinations at conveniently located places that are trusted, where community members have established relationships, and there is medical staff onsite to administer the vaccines and respond in the case of reactions. For example, one participant said: "*. . .more the merrier. That's more outlets for people to get vaccinated because some people don't like to go to hospitals, but they feel better getting it in the church or in the supermarket or in the pharmacy rather than the hospital itself*" (49-year-old female; community and social services occupation). Some mentioned considering a breadth of places where different population segments already access health or other services. Some interviewees also described ways of addressing other access barriers: ensuring that the cost of COVID-19 vaccines is covered for all individuals; providing public transportation (e.g., community shuttles), or partnering with rideshare companies to increase transportation access; and offering more flexible scheduling and vaccination site hours of operation.

When asked about their reactions to specific strategies to promote COVID-19 vaccines, interviewees discussed the perception that the vaccines could lead to greater freedom, including resuming social activities like visiting loved ones and traveling. Some interviewees felt that a monetary (e.g., gift card, cash incentive) or non-monetary (e.g., free meal for unhoused individuals) might incentivize vaccination uptake: "*. . .just being able to service the community or offer incentives. . . it can be like a monetary incentive. It can be like a one-time stipend or like the monetary incentive. Maybe for some, like the homeless community, a free meal*" (41-year-old female; community and social services occupation). However, a few had concerns around the underlying subtext and implications of providing an incentive for vaccination; for instance, the question of vaccine safety and efficacy if individuals had to be compensated to receive it. For example, one participant said: "*You're pimping out your body for money, and that's not okay*" (35-year-old female; business and financial operations occupation). Others said that those incentives would be ineffective in influencing behavior. Relatedly, some interviewees reported perceptions that the vaccines would eventually become mandatory, but many of them felt that such a requirement would cause them to seek workarounds, including alternative employment should their current employer mandated the vaccine.

## Discussion

The current paper reports qualitative findings from semi-structured interviews with Black Americans who showed low vaccine confidence as well as stakeholders representing different Black communities during the early stage of the COVID-19 vaccine roll-out. We documented low COVID-19 vaccination intentions (e.g., the need to wait and see) and specific concerns around the COVID-19 vaccines (e.g., about rapid development and side effects), which are consistent with similar qualitative research in racially/ethnically diverse populations (including Black Americans) and quantitative evidence that showed greater COVID-19 vaccine related mistrust in Black Americans compared to White Americans [11,12,24,25]. Consistent with prior literature on disparities related to the vaccination intentions for COVID-19 as well as seasonal influenza, HPV, and future HIV vaccines in Black Americans and other racial/ethnic minority groups [11,26–30], our data also showed substantial and multiple structural barriers to vaccination (e.g., transportation, financial, technology barriers that block vaccine access), as well as the negative impact of persistent systemic racism on vaccination intentions among Black Americans. Concerns of financial resources and costs have been documented for under-served communities for vaccine access as well as COVID-19 related prevention and coping [31,32]. Participants and stakeholders in the present study also identified system intervention strategies, from acknowledging systemic racism and improving the trustworthiness of institutions to preferred messaging contents, information sources and mode, and access points. While there were some novel ideas (e.g., discussing vaccination as empowerment), many of these strategies are consistent with the extant literature on strategies to increase COVID-19 vaccination update in Black communities and other racial/ethnic minority populations [11,12] as well as strategies to acknowledge medical mistrust in other health conditions such as HIV [13,17].

Based on our qualitative findings and quantitative findings reported elsewhere [19], we engaged community stakeholders to identify a set of public health messaging strategies and recommendations to help increase vaccination in Black communities. Many of the specific communication strategies and efforts recommended by the interview respondents were subsequently implemented to promote COVID-19 vaccination in minority and underserved communities. These strategies and recommendations also may be relevant going forward as boosters are disseminated—and may inform how community-based efforts can successfully increase vaccination intentions and decrease mistrust for future vaccination efforts (e.g., future HIV vaccines), for other health conditions, and around new medical technologies for future public health crises beyond the COVID-19 pandemic.

It is important to note that the strategies listed below should be embedded in long-term investment in capacity building and community engagement and based on a genuine effort to increase the trustworthiness of various institutions, including government, healthcare organizations, and public health systems. Public health and government officials should invest in creating long-term, equitable, and committed partnerships with underserved communities such as Black communities to increase their trustworthiness.

### Recommendations for increasing vaccine confidence

First, public health campaigns and messages to promote COVID-19 vaccines should acknowledge systemic racism and the justifiable multifaceted mistrust before providing information about the vaccines. It is important to understand and acknowledge mistrust as an understandable and justified response to personal experiences of racism and historical injustice. When exploring mistrust, it may be helpful to ask open-ended questions in a nonconfrontational, nonjudgmental way while providing accurate information.[5]

Second, public health messages and community strategies should be honest and transparent, using accessible and layperson language, and tailored toward the specific community based on research and understanding of the trusted information sources and the preferred contents and mode of communication. Our qualitative results suggest that people prefer a well-rounded discussion of the evidence so far, acknowledging both what we know and what we do not know about the vaccines and discussing both positive and negative results from clinical trials. This is a particularly imperative way to show transparency and honesty when conducting outreach to marginalized communities, such as people living with HIV, sexual and gender minorities, and immigrant communities.

Third, public health campaigns that promote the COVID-19 vaccines should provide opportunities for an open dialogue with scientists and health care providers who are trusted and credible sources for the relevant health information and can answer questions from community members. Formats such as online forums and Q&A sessions should be strongly considered. Our results emphasized the importance of leveraging trusted messengers and multiple dissemination mechanisms to share information transparently. Giving people the opportunities to ask questions is viewed as a way to build trust and increase buy-in for the vaccines.

Fourth, a tailored approach should be taken to create public health campaigns and messages to promote vaccination in specific communities. This approach means that the first step should be to recognize the diversity in Black communities across the country and do the necessary research to identify the trusted messengers and the preferred mode of information in the community. Specific strategies may be helpful, such as describing the vaccination as an empowerment process and emphasizing the individual's choice.

## Recommendations for increasing equity in vaccine access

The interviews were conducted before the vaccines were made available to all adults in the U.S. Therefore, issues with vaccine access were less discussed in early interviews, and problems with vaccine equity becomes increasingly striking as we concluded the interviews. However, our results offer suggestions for preferred vaccine access and strategies to resolve the most salient access issues. First, it is essential to provide multiple and different types of vaccine access points for a specific underserved community. Our findings show that people have mixed preferences for vaccination sites, as many are not comfortable getting vaccinated in medical settings due to medical providers' lack of trustworthiness, while others may feel most trusting towards their own doctors with whom they have a relationship. Therefore, offering more access points increase the chance of removing specific barriers.

Interviewees described a variety of structural barriers to vaccine access, from physical to digital barriers. Therefore, the process for vaccine access should be simplified and streamlined to maximize availability (e.g., allow for walk-in appointments). Different types of barriers should be removed, including lack of transportation (e.g., free ride, public transportation to vaccination sites) and difficulty navigating the online booking system. Ideally, the vaccination sites should meet people where they live and work to reduce access barriers. Our interviewees also recommended using the vaccination appointment as an opportunity to connect community members with health care and social service access.

## Limitations

A main limitation of the current study was the small sample size, and thus, we could not explore variation by meaningful subgroups (e.g., perspectives of healthcare workers vs. other types of participants). Relatedly, the qualitative sample of ALP participants was over-represented by those self-identified as female gender, while the sample of stakeholders was over-

represented by those self-identified as male gender. As such, we were unable to compare gender differences in the experience of barriers to vaccination as well as other factors that influence vaccination intentions. It will be important for future studies with larger sample sizes to document challenges to vaccination related attributed to racial identity as well as other vulnerabilities and marginalized identities such as socioeconomic status and sexual/gender minority status. Such data would help to guide the process of tailoring messages for specific subgroups affected by health inequities in general and during the pandemic. Another limitation is the low willingness among eligible ALP survey respondents to participate in the qualitative interviews (only about 30% of the eligible survey respondents completed the interviews). Because we invited those who endorsed the lowest vaccine intentions in the ALP survey to participate in the interviews, such individuals may have been more skeptical about vaccine-related research and thus may have decided not to participate. The qualitative sample appears to include more individuals with a healthcare and social service occupational background, potentially due to higher motivation for research participation in this group than individuals with other occupations.

## Conclusions

The current study presented qualitative data on drivers for Black Americans' low vaccination intentions and preferred strategies during the early stage of the COVID-19 vaccine roll-out in the U.S. A unique contribution of the current data and analysis is that they are focused on Black communities; in additional to stakeholders, the participants of this study were selected from a nationally representative sample of Black Americans based on their low vaccination intentions. These findings highlighted the importance of acknowledging and addressing mistrust and increasing efforts to improve equitable vaccine access as ways to improve vaccine confidence, vaccination intentions, and vaccination rates in Black communities. The recommended strategies may inform the roll-out of COVID-19 booster shots and improve the health care system's response not only to the current public health crisis but to future crises as well.

## Acknowledgments

We are grateful to all the study participants and members of the study's community stakeholder advisory committee for their guidance throughout this project.

## Author Contributions

**Conceptualization:** Lu Dong, Laura M. Bogart, James B. Aboagye, Bisola O. Ojikutu.

**Data curation:** Lu Dong, Laura M. Bogart, Priya Gandhi, Samantha Ryan.

**Formal analysis:** Lu Dong, Laura M. Bogart, Priya Gandhi.

**Funding acquisition:** Lu Dong, Laura M. Bogart.

**Investigation:** Laura M. Bogart, James B. Aboagye, Rosette Serwanga, Bisola O. Ojikutu.

**Methodology:** Lu Dong, Laura M. Bogart, Bisola O. Ojikutu.

**Project administration:** Lu Dong, Laura M. Bogart.

**Resources:** Laura M. Bogart.

**Supervision:** Lu Dong, Laura M. Bogart.

**Writing – original draft:** Lu Dong.

**Writing – review & editing:** Lu Dong, Laura M. Bogart, Priya Gandhi, James B. Aboagye, Samantha Ryan, Rosette Serwanga, Bisola O. Ojikutu.

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
