## [Decision Letter · Decision Letter 0]

20 Oct 2021

PONE-D-21-29026A Qualitative Study of COVID-19 Vaccine Intentions and Mistrust in Black Americans: Recommendations for Vaccine Dissemination and UptakePLOS ONE

Dear Dr. Dong,

Thank you for submitting your manuscript to PLOS ONE. After careful consideration, we feel that it has merit but does not fully meet PLOS ONE’s publication criteria as it currently stands. Therefore, we invite you to submit a revised version of the manuscript that addresses the points raised during the review process.

As the reviewer indicates, the introduction does well in contextualizing the important issues addressed, however the methods section needs more specifics, including citations, about the qualitative approach used and how the data were collected (e.g., by whom) and analyzed (more than providing quantitative metrics of reliability). (See journal submission guidelines: Qualitative research studies should be reported in accordance to the Consolidated criteria for reporting qualitative research (COREQ) checklist.) More consistent use of identiiers for quotations is also needed. In the bigger picture, as the reviewer suggests, some of the findings appear to be approached from an individual level that suggests a deficit model, and might be more profitably approached by engaging more fully with a structural perspective. Finally, the discussion section would be much improved by considering some of the rich literature used in the introduction and discussing the findings in the context of other research. The one refererence in the discussion is to a study from the same project; this is a missed opportunity to discuss your findings in the context of related research.

We look forward to receiving your revised manuscript.

Kind regards,

Peter A Newman, Ph.D

Academic Editor

PLOS ONE

2. When reporting the results of qualitative research, we suggest consulting the COREQ guidelines: http://intqhc.oxfordjournals.org/content/19/6/349. In this case, please consider including more information on the number of interviewers, their training and characteristics; and please provide the interview guide used.

3. Please provide additional details regarding participant consent. In the ethics statement in the Methods and online submission information, please ensure that you have specified: 1) whether the ethics committee approved the verbal/oral consent procedure, 2) why written consent could not be obtained, and 3) how verbal/oral consent was recorded. If your study included minors, please state whether you obtained consent from parents or guardians in these cases. If the need for consent was waived by the ethics committee, please include this information.

“We are grateful to all the study participants and members of the study’s community stakeholder advisory committee for their guidance throughout this project. Funding for this research was provided by gifts from RAND supporters and income from operations. The funders were not involved in the study design; data collection, analysis, and interpretation; or the preparation of this manuscript.”

“Funding for this research (awarded to LMB) was provided by gifts from RAND supporters and income from operations. The funders were not involved in the study design; data collection, analysis, and interpretation; or the preparation of this manuscript.”

Reviewers' comments:

Reviewer's Responses to Questions

**Comments to the Author**

1. Is the manuscript technically sound, and do the data support the conclusions?

Reviewer #1: Yes

Reviewer #2: Partly

2. Has the statistical analysis been performed appropriately and rigorously? 

Reviewer #1: N/A

Reviewer #2: N/A

3. Have the authors made all data underlying the findings in their manuscript fully available?

Reviewer #1: Yes

Reviewer #2: No

4. Is the manuscript presented in an intelligible fashion and written in standard English?

Reviewer #1: Yes

Reviewer #2: Yes

5. Review Comments to the Author

Reviewer #1: This manuscript provides a clear and detailed look at vaccine intentions and mistrust among Black Americans. One of the strengths of this manuscript is its capture of vaccine attitudes across time (e.g. when the vaccines were not yet available and when they had begun to be rolled out). The presentation of qualitative findings is well organized and well-supported in the manuscript. As the authors acknowledge, there is little discussion of intersectionality in either the ALP participant or stakeholder groups due to the lack of diversity in the small sample. This would be an interesting perspective to explore at a later date. Given the thorough examination of vaccine attitudes and intentions (related primarily to mistrust and access barriers) accompanied by actionable strategies, I would not hesitate to recommend this manuscript for publication.

Reviewer #2: This paper is writing about an important topic. I hope to make suggestions that would strengthen the presentation of the research.

Introduction

- Good overview of the issues. References literature that could be used to strengthen the discussion.

Materials and methods:

- Explain what the RAND American Life Panel is and explain what type of participants are recruitable from it. I noticed, for example, that there were quite a few health and social service professionals and that likely has an impact on the data that was gathered. Given the international audience of the journal, this data source is important to explain.

- There is no information provided about the procedures for the interview or who conducted the interviews. The latter point important, given the issues raised about mistrust and systemic racism.

- Provide information on ethics considerations or approval.

- Data analysis: This paper needs to locate the data analysis in a known model for qualitative data analysis and provide more detail about the procedure. This is important for establishing the trustworthiness (referencing qualitative research quality criteria) of the analysis.

Findings:

- It seems that the analysis focused on identifying data relevant to barriers and strategies (?facilitators) for vaccine acceptance. I would suggest that the literature review should have included a reference to the barriers/facilitators literature and this would have been the foundation for a clearer presentation of findings that aligned with those known barriers/facilitators. In addition, I believe it would shape the discussion of barriers as a description of problems in the system, rather than problems with the people trying to access the system. For example, what is identified a “convenience” barrier could also be identified as an availability barrier. The issue is not that people don’t want to be inconvenienced, the issue is that services are not available at times and locations that are accessible.

- The organization of findings into structural barriers and mistrust stemming from systemic racism is unclear. First, the concerns about privacy and security mentioned at the end of the structural section are not a structural barrier. They are another facet of mistrust that is being discussed in the next section. Second, mistrust, which I assume is evoked as another barrier, is locating the problem with the persons trying to access the system. I believe the barrier more accurately identified here is historical and systemic racism in healthcare and other systems that is the foundation for that mistrust. I suggest rethinking the organization of the findings, possibly by using an established framework for healthcare barriers/facilitators.

- As with the barriers, I think it is useful to explicitly position the strategies as system interventions that reduce the barriers located in the system.

- It is difficult for the reader to appreciate whether reported findings represent a majority or minority of participants and whether it represents different participants.

o The first issue is unclear because of the use of terms like “a few,” “a handful”; consider using language that is more explanatory (as has been done in several places) and using consistent terms across the findings section. I am aware that the authors may wish to avoid quantitative language in the description of findings; identifying an analytic approach would clarify the appropriate language for representing the data.

o The second issue is unclear because some quotes have a participant identified and others do not. In addition, there are sections of findings reported without any quotations to support them and no indication of whether they are representing things that were discussed by one, most, half or some portion of participants. All quotations should have identifying information with them so the reader has a context for the finding and knows they represent multiple voices. Ideally, all findings have quotations to substantiate them, to demonstrate the credibility of the analysis.

- Given the distribution of gender in the sample, were there any findings that suggested gendered experiences of barriers and facilitators? There seems to be a missed opportunity to consider gendered experiences in the analysis, discussion and limitations.

Tables:

- Table 3 and 4: These tables contain the quotations that would be useful to see in the text of the findings. The sources of quotations are not identified and that repeats the credibility problem in the finding section. A better use of these table would be to present the results of analysis that identified specific barriers and facilitators/strategies.

Discussion:

- End of page 17 there is a superscript 5 – is there a missing note 5?

- The discussion does not make sufficient use of extant literature to situate the findings of the study; the only research cited is previous papers published from the same study. There are useful sources referenced in the introduction. Literatures addressing Black communities and vaccine acceptance, or health care access more generally would be useful here. HIV testing? Diabetes management? HPV vaccines? Linking to these literatures will also strengthen the link between this study and future directions for public health.

- Limitations: is focused on who isn’t included but should also address how the findings of this study are relevant and useful as a contribution to knowledge.

Acknowledgements:

- Please explain “gifts from RAND supporters”; is this a donation? There may be standard language the funder suggests for crediting their contribution.

6. PLOS authors have the option to publish the peer review history of their article (what does this mean?). If published, this will include your full peer review and any attached files.

Reviewer #1: **Yes: **Kate Allan

Reviewer #2: No

---

## [Author Response · Author response to Decision Letter 0]

18 Jan 2022

We are grateful for the helpful and incisive reviews and for this opportunity to submit a revised version. We have carefully considered the reviewers’ helpful comments and have revised the paper in accordance. Our response and the changes made to the manuscript are detailed below (marked with * after each point).

Editor’s Comments: 

1. As the reviewer indicates, the introduction does well in contextualizing the important issues addressed, however the methods section needs more specifics, including citations, about the qualitative approach used and how the data were collected (e.g., by whom) and analyzed (more than providing quantitative metrics of reliability). (See journal submission guidelines: Qualitative research studies should be reported in accordance to the Consolidated criteria for reporting qualitative research (COREQ) checklist.) 

*We have provided additional details according to the COREQ guidelines. The COREQ checklist is uploaded as part of the resubmission for review. 

2. More consistent use of identifiers for quotations is also needed. In the bigger picture, as the reviewer suggests, some of the findings appear to be approached from an individual level that suggests a deficit model, and might be more profitably approached by engaging more fully with a structural perspective. 

*We have ensured that all quotes (regardless of phrases or full quotes) have identifiers. We have also edited the results and discussion according to Reviewer 2’s suggestions to engage a structural perspective. 

3. Finally, the discussion section would be much improved by considering some of the rich literature used in the introduction and discussing the findings in the context of other research. The one reference in the discussion is to a study from the same project; this is a missed opportunity to discuss your findings in the context of related research.

*We have expanded the discussion section to include a discussion of findings in the context of existing research (page 19, paragraph 2). In addition, we have updated the literature review to reflect most current national data on vaccination rates in Black Americans as well as key literature published since our submission (page 3, paragraph 1; page 4, paragraph 3 to page 5 paragraph 1).

Reviewer #1: 

1. This manuscript provides a clear and detailed look at vaccine intentions and mistrust among Black Americans. One of the strengths of this manuscript is its capture of vaccine attitudes across time (e.g., when the vaccines were not yet available and when they had begun to be rolled out). The presentation of qualitative findings is well organized and well-supported in the manuscript. As the authors acknowledge, there is little discussion of intersectionality in either the ALP participant or stakeholder groups due to the lack of diversity in the small sample. This would be an interesting perspective to explore at a later date. Given the thorough examination of vaccine attitudes and intentions (related primarily to mistrust and access barriers) accompanied by actionable strategies, I would not hesitate to recommend this manuscript for publication.

*Thank you very much for this positive feedback. 

Reviewer #2: 

This paper is writing about an important topic. I hope to make suggestions that would strengthen the presentation of the research.

Introduction:

1. Good overview of the issues. References literature that could be used to strengthen the discussion.

*Thank you. we have strengthened the discussion section following the reviewer’s suggestions. 

Materials and methods:

2. Explain what the RAND American Life Panel is and explain what type of participants are recruitable from it. I noticed, for example, that there were quite a few health and social service professionals and that likely has an impact on the data that was gathered. Given the international audience of the journal, this data source is important to explain.

*We have added more information describing the American Life Panel on page 5 end of the page to page 6 paragraph 1: “Participants were drawn from the RAND American Life Panel (ALP), a nationally representative internet panel of U.S. adults aged 18 and older. The ALP currently has over 3000 active members, who have been recruited from several other surveys and directly for the panel using multiple modes (e.g., in-person, telephone, and mail) and probability-based sampling methods, including address-based samples and telephone (random-digit dial) samples. Full details of the ALP’s methodology are available in the technical documentation (1).”

Individuals who expressed high mistrust and low vaccination intention were invited to participate in the qualitative interviews. It is possible that those who are in health and social service professions were interested in participating. We have noted this as a limitation in the discussion section (see page 23 paragraph 1). 

3. There is no information provided about the procedures for the interview or who conducted the interviews. The latter point important, given the issues raised about mistrust and systemic racism.

*We provided this information, per COREQ checklist (see uploaded checklist for review), on page 8 paragraph 2: “Two interviewers (a mixed-race woman of African and European descent [SR] and a woman of South Asian descent [PG]) completed training provided by a White female psychologist (LMB) and an Asian female psychologist (LD)…. The community advisory board and all co-authors reviewed and provided feedback on the interview guide.”

4. Provide information on ethics considerations or approval.

*We provided this information on page 6 paragraph 2: “RAND’s Institutional Review Board approved the study design and protocol.”

5. Data analysis: This paper needs to locate the data analysis in a known model for qualitative data analysis and provide more detail about the procedure. This is important for establishing the trustworthiness (referencing qualitative research quality criteria) of the analysis.

*Thank you for this suggestion. We have now provided more details about the qualitative data analysis on page 9 paragraph 2: “We followed standard procedures for direct content analysis (2), in which we developed the initial codebook based on existing research and from directly interpreting the meaning of the interview transcripts. Specifically, two experienced team members (LMB and LD) read all transcripts and developed a draft codebook based on the research questions, relevant literature, and the interpretation of the narratives. Two coders (LD and PG) coded six transcripts to test the codebook and make necessary updates through iterations.” 

Findings:

6. It seems that the analysis focused on identifying data relevant to barriers and strategies (?facilitators) for vaccine acceptance. I would suggest that the literature review should have included a reference to the barriers/facilitators literature and this would have been the foundation for a clearer presentation of findings that aligned with those known barriers/facilitators. In addition, I believe it would shape the discussion of barriers as a description of problems in the system, rather than problems with the people trying to access the system. For example, what is identified a “convenience” barrier could also be identified as an availability barrier. The issue is not that people don’t want to be inconvenienced, the issue is that services are not available at times and locations that are accessible.

*In response to this reviewer’s suggestion, we have added more relevant literature on barriers/facilitators for COVID vaccines in underserved populations, including papers published more recently (since our original submission). We have also explicitly discussed vaccine access issue as a barrier. These changes are made to the introduction on pages 3-5.

Following the reviewer’s suggestion, we have renamed “convenience” to “availability” throughout the manuscript. 

7. The organization of findings into structural barriers and mistrust stemming from systemic racism is unclear. First, the concerns about privacy and security mentioned at the end of the structural section are not a structural barrier. They are another facet of mistrust that is being discussed in the next section. Second, mistrust, which I assume is evoked as another barrier, is locating the problem with the persons trying to access the system. I believe the barrier more accurately identified here is historical and systemic racism in healthcare and other systems that is the foundation for that mistrust. I suggest rethinking the organization of the findings, possibly by using an established framework for healthcare barriers/facilitators.

*Following the reviewer’s suggestions, we have made the following changes: 

1) We have now integrated the barrier about privacy and security concerns into mistrust section (now page 16, paragraph 1). We renamed the mistrust section as systemic racism as a barrier that leads to mistrust, and discussed mistrust as a consequence of systemic racism. We have re-organized the paragraphs about systemic racism and mistrust on page 13-16. 

2) We have considered using an established framework for healthcare barriers/facilitators, but ultimately decided not to use one because there is not a good fit. Most of these frameworks are from implementation science and focus on healthcare providers/systems and organizations, and barriers at multiple levels. In addition, from an analytic perspective, we did not use a framework, but instead allowed the themes to emerge from the narratives. 

8. As with the barriers, I think it is useful to explicitly position the strategies as system interventions that reduce the barriers located in the system.

*Thank you for this suggestion. We have renamed “strategies” as “system-level interventions” in the results and discussion sections as well as Table 4.

9. It is difficult for the reader to appreciate whether reported findings represent a majority or minority of participants and whether it represents different participants.

a) The first issue is unclear because of the use of terms like “a few,” “a handful”; consider using language that is more explanatory (as has been done in several places) and using consistent terms across the findings section. I am aware that the authors may wish to avoid quantitative language in the description of findings; identifying an analytic approach would clarify the appropriate language for representing the data.

*Thank you for this suggestion. We have now updated the results section using the following categories (page 10, paragraph 1): “Following prior research (3), we used the following categories to indicate the relative frequency of each code: most or almost all (61%-100%), some (21%-60%), and a few (1%-20%). Given that the interviews were semi-structured and used open-ended questions, providing exact percentages for codes could be potentially misleading as a lack of discussion of a theme did not mean agreeing or disagreeing with the idea.”

b) The second issue is unclear because some quotes have a participant identified and others do not. In addition, there are sections of findings reported without any quotations to support them and no indication of whether they are representing things that were discussed by one, most, half or some portion of participants. All quotations should have identifying information with them so the reader has a context for the finding and knows they represent multiple voices. Ideally, all findings have quotations to substantiate them, to demonstrate the credibility of the analysis.

*Thank you for the opportunity to clarify. We included identifiers for all quotes. Note that there were a few phrases that we quoted the exact phrasing used by participants in our description of results without including identifiers because those were not formally listed as quotes. We recognize that this can be confusing, so we have now added identifiers to all quotations. 

We also clarify that all findings do have quotations to substantiate them. For the section on “COVID-19 vaccination intentions,” we directly included all the relevant quotes in the text because it is manageable to do so. For other sections (i.e., barriers, system intervention strategies), we included the majority of relevant quotes in Tables 3 and 4 because we find the table format is efficient in presenting quotes to illustrate each sub-finding and by interviewee types (ALP participants vs. stakeholders). 

10. Given the distribution of gender in the sample, were there any findings that suggested gendered experiences of barriers and facilitators? There seems to be a missed opportunity to consider gendered experiences in the analysis, discussion and limitations.

*In our sample, most ALP participants were women (19 out of 24 interviewees) and most stakeholders are men (4 out of 5). We do not have sufficient data to do a formal comparison. We have added this as a limitation in the discussion (page 23 paragraph 1). 

Tables:

11. Table 3 and 4: These tables contain the quotations that would be useful to see in the text of the findings. The sources of quotations are not identified and that repeats the credibility problem in the finding section. A better use of these table would be to present the results of analysis that identified specific barriers and facilitators/strategies.

*We included an identifier (age, gender, occupation) at the end of each quote, in Tables 3 and 4 as well as when quotes are directly included in the results section. 

Given that there are many quotes that are relevant to present and illustrate each barrier and facilitator/strategy, we prefer to retain the format of including the majority of quotes in the Table format. The table format is also more efficient to list quotes from both ALP participants and stakeholders. However, If the editor prefers, we can integrate the quotes into the text. 

Discussion:

12. End of page 17 there is a superscript 5 – is there a missing note 5?

*Thank you for spotting this. We have removed this and added the correct citation. 

13. The discussion does not make sufficient use of extant literature to situate the findings of the study; the only research cited is previous papers published from the same study. There are useful sources referenced in the introduction. Literatures addressing Black communities and vaccine acceptance, or health care access more generally would be useful here. HIV testing? Diabetes management? HPV vaccines? Linking to these literatures will also strengthen the link between this study and future directions for public health.

*Thank you for this suggestion. We have now strengthened the discussion section in terms of better situating the findings in the extant literature (page 19, paragraph 2). We have also added more citations on health disparities in other types of vaccinations that the current results were in line with and could be informative for addressing these other health-related issues. 

14. Limitations: is focused on who isn’t included but should also address how the findings of this study are relevant and useful as a contribution to knowledge.

*We have described the contribution of the study in the conclusion section (page 23, paragraph 1). 

Acknowledgements:

15. Please explain “gifts from RAND supporters”; is this a donation? There may be standard language the funder suggests for crediting their contribution.

*Yes, this means donation from supporters, and this is the standard language for acknowledgement. 

In summary, thank you for the helpful feedback and the opportunity to submit this revised version of our manuscript. We believe these additional changes have resulted in an improved paper. We would, of course, welcome further guidance on improving the manuscript.

Yours sincerely,

Authors

References cited

1. Pollard MS, Baird MD. The RAND American Life Panel: Technical Description. Santa Monica, CA: RAND Corporation; 2017.

2. Hsieh HF, Shannon SE. Three approaches to qualitative content analysis. Qual Health Res. 2005;15(9):1277-88.

3. Bogart LM, Cowgill BO, Sharma AJ, Uyeda K, Sticklor LA, Alijewicz KE, et al. Parental and home environmental facilitators of sugar-sweetened beverage consumption among overweight and obese Latino youth. Acad Pediatr. 2013;13(4):348-55.

---

## [Decision Letter · Decision Letter 1]

1 Mar 2022

PONE-D-21-29026R1A Qualitative Study of COVID-19 Vaccine Intentions and Mistrust in Black Americans: Recommendations for Vaccine Dissemination and UptakePLOS ONE

Dear Dr. Dong,

Thank you for submitting your manuscript to PLOS ONE. After careful consideration, we feel that it has merit but does not fully meet PLOS ONE’s publication criteria as it currently stands. Therefore, we invite you to submit a revised version of the manuscript that addresses the points raised during the review process.

 The reviewer noted that their comments were largely addressed and the manuscript much improved. Kindly respond to the minor revisions requested, at which point the manuscript can be accepted for publication.

We look forward to receiving your revised manuscript.

Kind regards,

Peter A Newman, Ph.D

Academic Editor

PLOS ONE

Journal Requirements:

Reviewers' comments:

Reviewer's Responses to Questions

**Comments to the Author**

1. If the authors have adequately addressed your comments raised in a previous round of review and you feel that this manuscript is now acceptable for publication, you may indicate that here to bypass the “Comments to the Author” section, enter your conflict of interest statement in the “Confidential to Editor” section, and submit your "Accept" recommendation.

Reviewer #2: (No Response)

2. Is the manuscript technically sound, and do the data support the conclusions?

Reviewer #2: Yes

3. Has the statistical analysis been performed appropriately and rigorously? 

Reviewer #2: N/A

4. Have the authors made all data underlying the findings in their manuscript fully available?

Reviewer #2: No

5. Is the manuscript presented in an intelligible fashion and written in standard English?

Reviewer #2: Yes

6. Review Comments to the Author

Reviewer #2: The revisions have strengthened the paper. Some suggestions that might strengthen it further:

I would encourage the authors to consider whether some of the in-text quotations are too truncated, making them poor representations of the ideas they are claimed to represent. Examples are on page 14 "looking for profit" and page 15 "fear and anxiety".

In-text quotations disappear after page 17. There are good quotations in the table that speak to the content but it might be nice to include something in the latter half of the findings as it makes the findings section unbalanced.

I would strongly recommending reconsidering the claim on page 22: “our results also indicated challenges that are attributed to racial identity and other vulnerabilities and marginalized identities such as socioeconomic status and sexual/gender identities.” There was no gender or sexual identity analysis visible in the results section, therefore, this reads as speaking beyond the results. The point is a good one but would be better identified as something to be pursued in future research. The findings do reference financial barriers and that could be discussed in the discussion, with support of relevant literature.

7. PLOS authors have the option to publish the peer review history of their article (what does this mean?). If published, this will include your full peer review and any attached files.

Reviewer #2: **Yes: **Charmaine C. Williams, PhD

---

## [Author Response · Author response to Decision Letter 1]

14 Apr 2022

Reviewer #2: 

The revisions have strengthened the paper. Some suggestions that might strengthen it further:

1. I would encourage the authors to consider whether some of the in-text quotations are too truncated, making them poor representations of the ideas they are claimed to represent. Examples are on page 14 "looking for profit" and page 15 "fear and anxiety".

*Thank you for the additional suggestions. We have removed these short, truncated in-text quotations, and changed them into full quotations as appropriate (please see pages 13-16, marked version). 

2. In-text quotations disappear after page 17. There are good quotations in the table that speak to the content, but it might be nice to include something in the latter half of the findings as it makes the findings section unbalanced.

*We have included selected exemplar quotes in text on pages 17-20 (marked version). 

3. I would strongly recommend reconsidering the claim on page 22: “our results also indicated challenges that are attributed to racial identity and other vulnerabilities and marginalized identities such as socioeconomic status and sexual/gender identities.” There was no gender or sexual identity analysis visible in the results section, therefore, this reads as speaking beyond the results. The point is a good one but would be better identified as something to be pursued in future research. The findings do reference financial barriers and that could be discussed in the discussion, with support of relevant literature.

*We have revised the sentence (on gender or sexual identity) to discuss a future research direction (page 24, paragraph 3, marked version). In addition, we added a discussion point on financial barriers (page 21, paragraph 1, marked version).

---

## [Editor Report · Decision Letter 2]

21 Apr 2022

A Qualitative Study of COVID-19 Vaccine Intentions and Mistrust in Black Americans: Recommendations for Vaccine Dissemination and Uptake

PONE-D-21-29026R2

Dear Dr. Dong,

Thank you for your receptiveness and revisions in response to the reviewer's comments. We’re pleased to inform you that your manuscript has been judged scientifically suitable for publication and will be formally accepted for publication once it meets all outstanding technical requirements.

Kind regards,

Peter A. Newman, Ph.D

Academic Editor

PLOS ONE
---

## [Editor Report · Acceptance letter]

25 Apr 2022

PONE-D-21-29026R2 

A Qualitative Study of COVID-19 Vaccine Intentions and Mistrust in Black Americans: Recommendations for Vaccine Dissemination and Uptake 

Dear Dr. Dong:

I'm pleased to inform you that your manuscript has been deemed suitable for publication in PLOS ONE. Congratulations! Your manuscript is now with our production department. 

Kind regards, 

on behalf of

Dr. Peter A. Newman 

Academic Editor

PLOS ONE